# Modulation of riboflavin biosynthesis and utilization in mycobacteria

Melissa D. Chengalroyen,[1] Carolina Mehaffy,[2] Megan Lucas,[2] Niel Bauer,[2] Mabule L. Raphela,[1] Nurudeen Oketade,[2] Digby F. Warner,[1,3] Deborah A. Lewinsohn,[4] David M. Lewinsohn,[4,5] Karen M. Dobos,[2] Valerie Mizrahi[1,3]

**ABSTRACT** Riboflavin (vitamin $B_2$) is the precursor of the flavin coenzymes, FAD and FMN, which play a central role in cellular redox metabolism. While humans must obtain riboflavin from dietary sources, certain microbes, including *Mycobacterium tuberculosis* (Mtb), can biosynthesize riboflavin *de novo*. Riboflavin precursors have also been implicated in the activation of mucosal-associated invariant T (MAIT) cells which recognize metabolites derived from the riboflavin biosynthesis pathway complexed to the MHC-I-like molecule, MR1. To investigate the biosynthesis and function of riboflavin and its pathway intermediates in mycobacterial metabolism and physiology, we constructed conditional knockdowns (hypomorphs) in riboflavin biosynthesis and utilization genes in *Mycobacterium smegmatis* (Msm) and Mtb by inducible CRISPR interference. Using this comprehensive panel of hypomorphs, we analyzed the impact of gene silencing on viability, on the transcription of (other) riboflavin pathway genes, on the levels of the pathway proteins, and on riboflavin itself. Our results revealed that (i) despite lacking a canonical transporter, both Msm and Mtb assimilate exogenous riboflavin when supplied at high concentration; (ii) there is functional redundancy in lumazine synthase activity in Msm; (iii) silencing of *ribA2* or *ribF* is profoundly bactericidal in Mtb; and (iv) in Msm, *ribA2* silencing results in concomitant knockdown of other pathway genes coupled with RibA2 and riboflavin depletion and is also bactericidal. In addition to their use in genetic validation of potential drug targets for tuberculosis, this collection of hypomorphs provides a useful resource for future studies investigating the role of pathway intermediates in MAIT cell recognition of mycobacteria.

**IMPORTANCE** The pathway for biosynthesis and utilization of riboflavin, precursor of the essential coenzymes, FMN and FAD, is of particular interest in the flavin-rich pathogen, *Mycobacterium tuberculosis* (Mtb), for two important reasons: (i) the pathway includes potential tuberculosis (TB) drug targets and (ii) intermediates from the riboflavin biosynthesis pathway provide ligands for mucosal-associated invariant T (MAIT) cells, which have been implicated in TB pathogenesis. However, the riboflavin pathway is poorly understood in mycobacteria, which lack canonical mechanisms to transport this vitamin and to regulate flavin coenzyme homeostasis. By conditionally disrupting each step of the pathway and assessing the impact on mycobacterial viability and on the levels of the pathway proteins as well as riboflavin, our work provides genetic validation of the riboflavin pathway as a target for TB drug discovery and offers a resource for further exploring the association between riboflavin biosynthesis, MAIT cell activation, and TB infection and disease.

**KEYWORDS** tuberculosis, *Mycobacterium*, FAD, FMN, drug discovery

Riboflavin (RF; vitamin $B_2$) is an essential micronutrient required by all life forms. Serving as a precursor for the coenzymes FMN and FAD, this vitamin is the

Address correspondence to Melissa D. Chengalroyen, mel.chengalroyen@uct.ac.za, or Valerie Mizrahi, Valerie.mizrahi@uct.ac.za.

The authors declare no conflict of interest.

See the funding table on p. 16.

10.1128/spectrum.03207-23 **1**

foundation of the entire flavoproteome which comprises 2%–3% of all known enzymes (1). Every living organism uses FMN and FAD to mediate one- and two-electron oxidation/reduction reactions in metabolic pathways (1). Humans, however, are unable to biosynthesize RF and must obtain this vitamin from dietary sources (2). RF proto-trophic microbes synthesize this vitamin *de novo* from ribulose-5-phosphate and GTP via a conserved pathway with bacteria frequently possessing multiple paralogs of genes encoding RF biosynthetic pathway enzymes (3, 4). The *Mycobacterium tuberculosis* (Mtb) genome encodes ~140 flavoproteins which, together, constitute a comparatively large flavoproteome that gives effect to this organism's "flavin-intensive lifestyle" (5). Mycobacteria also biosynthesize and utilize the deazaflavins, $F_0$ and $F_{420}$, which are derived from the flavin biosynthesis pathway intermediate, 5-amino-6-ribityla-mino-2,4(1H,3H)-pyrimidinedione (5-A-RU) (6, 7). While dispensable for Mtb growth *in vitro* (8), $F_{420}$ has been implicated in non-replicating persistence of Mtb and in response to oxidative, nitrosative, and antibiotic-mediated stress (6). Interest in the role of deazaflavoproteins in Mtb metabolism and physiology was heightened significantly by the discovery that the deazaflavin-dependent nitroreductase, Ddn, activates, by hydride transfer from $H_2F_{420}$ to the nitroimidazole moiety, the new nitroimidazole antitubercular prodrugs, pretomanid (9) and delamanid (10), which were recently approved for the treatment of drug-resistant TB (11). The RF biosynthesis pathway has also attracted interest as a potential target for antimicrobial drug discovery, especially for pathogens such as Mtb that lack a canonical RF transporter (12–14). The bifunctional GTP cyclohy-drolase II/3,4-dihydroxy-2-butanone-4-phosphate (3,4-DHBP) synthase, RibA2, lumazine synthase, RibH, and RF synthase, RibC, are considered promising antimicrobial targets given their absence in humans (15–17). The crystal structures of Mtb RibA2 (18) and RibH (19) have been solved, and inhibitors of the RF-utilizing enzyme, RibF, designed (20).

In addition to its role as the flavin coenzyme precursor, RF has also been implicated in the activation of mucosal-associated invariant T (MAIT) cells (21). These specialized T-cells bridge innate and adaptive immunity by serving as rapid responders to invasive bacterial infections akin to innate immune cells and attain effector memory like adaptive immune cells (22, 23). MAIT cells specifically recognize antigens complexed to the MHC-I-like molecule, MR1. Unlike antigenic peptides or lipids presented to conventional T-cells, MR1 presents microbial metabolites derived from the RF biosynthetic pathway to activate MAIT cells, which leads to the production of the inflammatory cytokines, IFN-γ, TNF-α, and IL-17 to control infection in conjunction with granzyme B and perforin, which destroy the infected cells (24). Since humans cannot manufacture RF precursors, these antigens are highly specific for invasion by pathogens including Mtb, *Helicobacter pylori*, *Salmonella typhimurium*, *Escherichia coli*, and *Staphylococcus aureus*, which have intact RF biosynthesis pathways (21, 25). 5-A-RU has been identified as the central intermediate in modulating MAIT cell activity (26). This pathway intermediate can yield products that are either activators or antagonists of MAIT cell function, depending on the molecule with which 5-AR-U interacts, although it has no capacity to bind MR1 itself (27). MAIT cells have been shown to facilitate the control of *Klebsiella pneumoniae* (28), *Francisella tularensis* (29), and *Mycobacterium bovis* BCG in mice (30, 31). However, mice deficient in MR1 do not exhibit a defect in Mtb control (32) and vaccination with the MR1 ligand, 5-(2-oxopropylideneamino)-6-D-ribitylaminouracil (5-OP-RU) was not effective (32, 33). Interestingly, 5-OP-RU treatment of mice with chronic Mtb infection reduced the bacillary load (32), but this effect was not recapitulated in non-human primates (34). Therefore, whether and how the administration of MAIT-activating ligands such as 5-OP-RU might impact upon TB disease progression in humans remain unresolved.

Against this background, we sought to investigate RF biosynthesis and utilization in Mtb as well as the model organism, *Mycobacterium smegmatis* (Msm). To this end, we constructed a set of conditional hypomorphs in RF biosynthesis and utilization genes by inducible CRISPR interference (CRISPRi) (35) and determined the impact of RF pathway disruption on expression of the pathway genes, and also, in Msm, on the levels of the encoded proteins as well as RF. We confirm that both organisms can transport and

assimilate exogenous RF when supplied at a high concentration. We also show that, while Msm has two genes that encode RibH activity, Mtb has only one, and that silencing of *ribA2*, *ribG*, *ribH,* or *ribF* is bactericidal in Mtb *in vitro*. In addition to genetically validating RF biosynthesis and utilization as potential TB drug targets, the collection of hypomorphs described in this paper provides a potentially valuable resource for future investigation of the association between RF pathway disruption in mycobacteria and MAIT cell activation.

## RESULTS

### Pathway for RF biosynthesis and utilization in mycobacteria

The genomic organization and proposed pathway for RF biosynthesis and utilization in mycobacteria are shown in Fig. 1. The genes, the functions of their encoded proteins, and their associated essentiality and vulnerability data are summarized in Table S1. For simplicity, the same nomenclature is used for the Mtb and Msm pathway genes. The biosynthetic pathway involves the conversion of GTP and ribulose 5-phosphate to RF via a stepwise process catalyzed by five enzymes: a bifunctional GTP cyclohydrolase II/DHBP synthase (RibA2), RF deaminase/5-amino-6-(5-phosphoribosylamino) uracil reductase (RibG), an unknown phosphatase, lumazine synthase (RibH), and RF synthase (RibC).

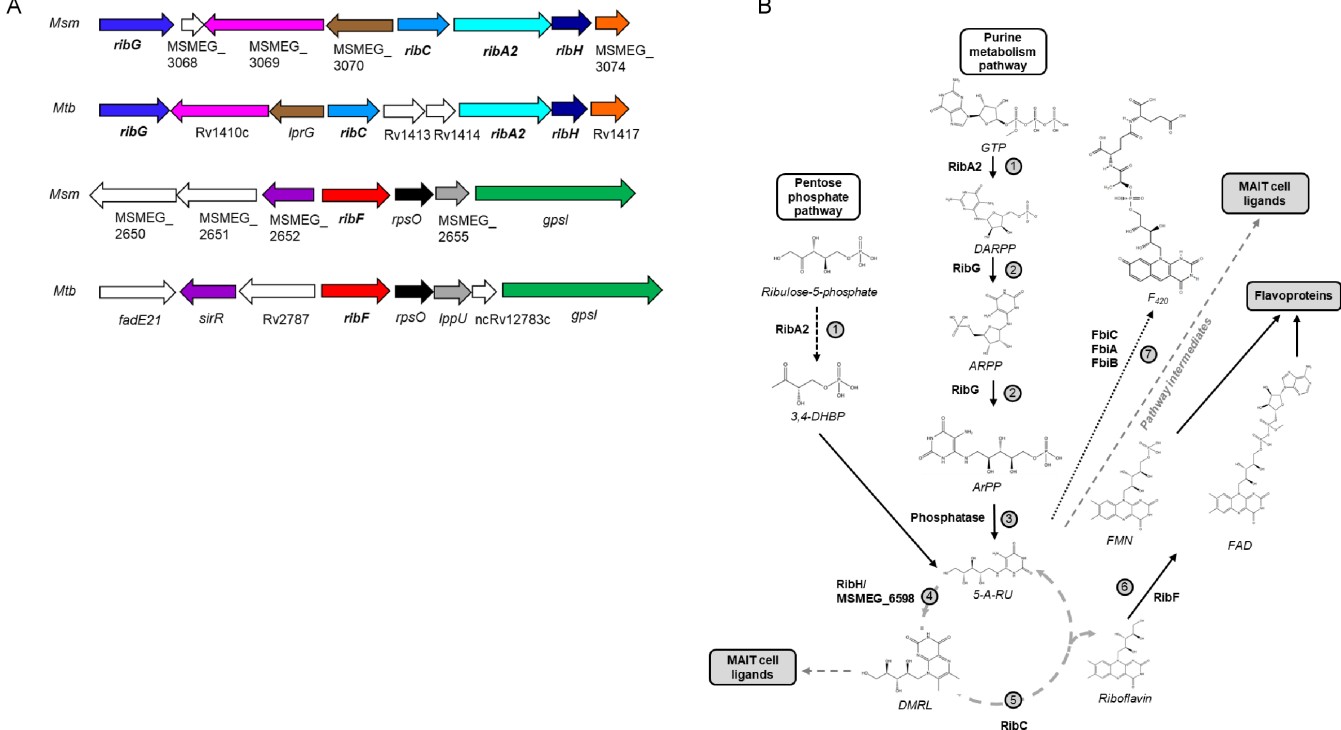

**FIG 1** Proposed RF pathway in mycobacteria. (A) Genomic context of RF pathway genes in Msm and Mtb. Homologs are shown in the same color. (B) Step 1. The bifunctional GTP cyclohydrolase II/DHBP synthase, RibA2 catalyzes the first steps of the pathway by converting two molecules of ribulose-5-P, emanating from purine metabolism (solid black arrows) and one GTP molecule derived from the pentose phosphate route (dashed black arrows), to their respective products, 3,4-dihydroxy-2-butanone 4-phosphate (3,4-DHBP) and 2,5-diamino-6-(5-phospho-D-ribosylamino)-pyrimidin-4(3H)-one (DARPP) (18). Steps 2 and 3. DARPP is deaminated by the bifunctional RF deaminase/5-amino-6-(5-phosphoribosylamino) uracil reductase RibG to 5-amino-6-(5′-phospho-ribitylamino)-uracil (ARPP). RibG also reduces and opens the ring of the ribofuranose group of ARPP to form 5-amino-6-ribitylamino-2,4(1H,3H)-pyrimidinedione-5-phosphate (ArPP). ArPP is dephosphorylated by an unknown phosphatase to produce 5-amino-6-D-ribitylaminouracil (5-A-RU). Step 4. The products from the two pathways, 3,4-DHBP and 5-A-RU, converge and are condensed by lumazine synthase, RibH, yielding 6,7-dimethyl-8-ribityllumazine (DMRL). Step 5. The RF synthase, RibC, through a dismutation reaction, converts two molecules of DMRL to RF and 5-A-RU (which feeds back into the cycle). Step 6. RF is converted to FMN and FAD through by the bifunctional kinase/FAD synthetase, RibF (36). Step 7. Through distinct enzymatic reactions mediated by FbiC, FbiB, and FbiA, 5-A-RU can also be converted to $F_{420}$ (37). DMRL and its derivatives can act as MAIT cell activators. Additionally, 5-A-RU also has the capacity to interact with molecules from specific pathways to produce MAIT cell modulators (gray dashed line). Msm and Mtb also possess a monofunctional GTP cyclohydrolase, RibA1.

RF is then converted to the coenzymes, FMN and FAD, by the bifunctional kinase/ FAD synthase, RibF.

The complement and genomic organization of RF biosynthesis genes vary considerably among bacteria (3). In *Bacillus subtilis*, the genes are organized on a five-gene operon (38), whereas in *E. coli*, they are dispersed across the genome (39–41). The genomic organization is broadly conserved between Mtb and Msm with some differences (Fig. 1). RT-PCR analysis suggested that the contiguous *ribC*, *ribA2*, and *ribH* genes form a single transcriptional unit in Msm (Fig. S1). The Mtb homologs were also found to constitute a single transcriptional unit (the "*rib*" operon), but, in this case, the *ribC* and *ribA2* genes are separated by *Rv1413* and *Rv1414*, which encode conserved hypothetical proteins of unknown function with some homology to pyridoxal 5-phosphate-dependent enzymes (Fig. 1). In both mycobacterial species, *ribG* is located upstream of the *rib* operon, and separated from *ribC* by two (Mtb) or three (Msm) genes. As observed in other bacteria (3), Mtb and Msm show paralogous expansion of certain RF biosynthesis genes (Table S1). In addition to *ribA2*, both Msm and Mtb also carry *ribA1*, which encodes a putative standalone GTP cyclohydrolase II, and Msm carries a second *ribH* homolog, MSMEG_6598. In accordance with other reports (13), we could not identify a canonical FMN riboswitch (*FMN* element) (42, 43) in the 5′ untranslated region of *ribG* or *ribC* in either organism.

## Construction of RF pathway hypomorphs in mycobacteria

MAIT cells predominantly recognize metabolites of microbial origin from the RF biosynthesis pathway. However, the intermediates within this pathway can act as either T-cell activators or antagonists depending on the downstream substrates with which they interact (27). To investigate this in greater detail, we constructed a panel of hypomorphs in Msm and Mtb by inducible CRISPRi to enable conditional silencing at each step in the pathway. We reasoned that the tunable nature of this system (35, 44) might allow the levels of pathway intermediates up- and down-stream of the targeted step to be modulated with potential impact on MAIT cell activation or antagonism. The hypomorphs would also allow the impact of target gene silencing on mycobacterial viability to be compared between steps in the pathway, and across the two organisms, which, in turn, could inform early-stage drug discovery. For cases in which two different RF biosynthesis genes were annotated as encoding the same function (e.g., *ribA1* and *ribA2*), we focused initially on those that clustered in the same genomic locus, i.e., within or proximal to the *rib* operon (Fig. 1). For each gene, 5–15 sgRNAs with varying predicted efficiencies of knockdown were used (Fig. S2). All resulting hypomorphs were screened for ATc responsiveness over a range of ATc concentrations (0.1–200 ng/mL; data not shown). For each gene, the most ATc-responsive hypomorph (or the one with the lowest PAM score, in the case of *ribC*) was selected for further characterization (Fig. S2). To examine the effect of simultaneously silencing two genes in the pathway, we constructed duplexed hypomorphs of Msm in which *ribA2* silencing was coupled with silencing of *ribG* or *ribF*.

## Silencing of RF pathway genes differentially impacts mycobacterial viability

The hypomorphs were analyzed in terms of growth on solid media alongside vector-only controls (Fig. 2). Overall, the Mtb hypomorphs were more sensitive to ATc-mediated growth inhibition than their Msm counterparts, consistent with their greater vulnerabilities (Table S1). Accordingly, Msm was least vulnerable to *ribG* or *ribC* knockdown with hypomorphs in these genes showing little or no phenotype in this assay. Both duplexed hypomorphs showed more profound ATc-dependent growth inhibition than either of the corresponding single-gene hypomorphs, demonstrating negligible growth in the presence of ATc.

Broadly similar phenotypes were observed when the hypomorphs were cultured in liquid media using turbidity ($OD_{600}$) and viability (CFU enumeration) to monitor the impact of ATc treatment (Fig. S3 and S4). As a further control for potential effects of the

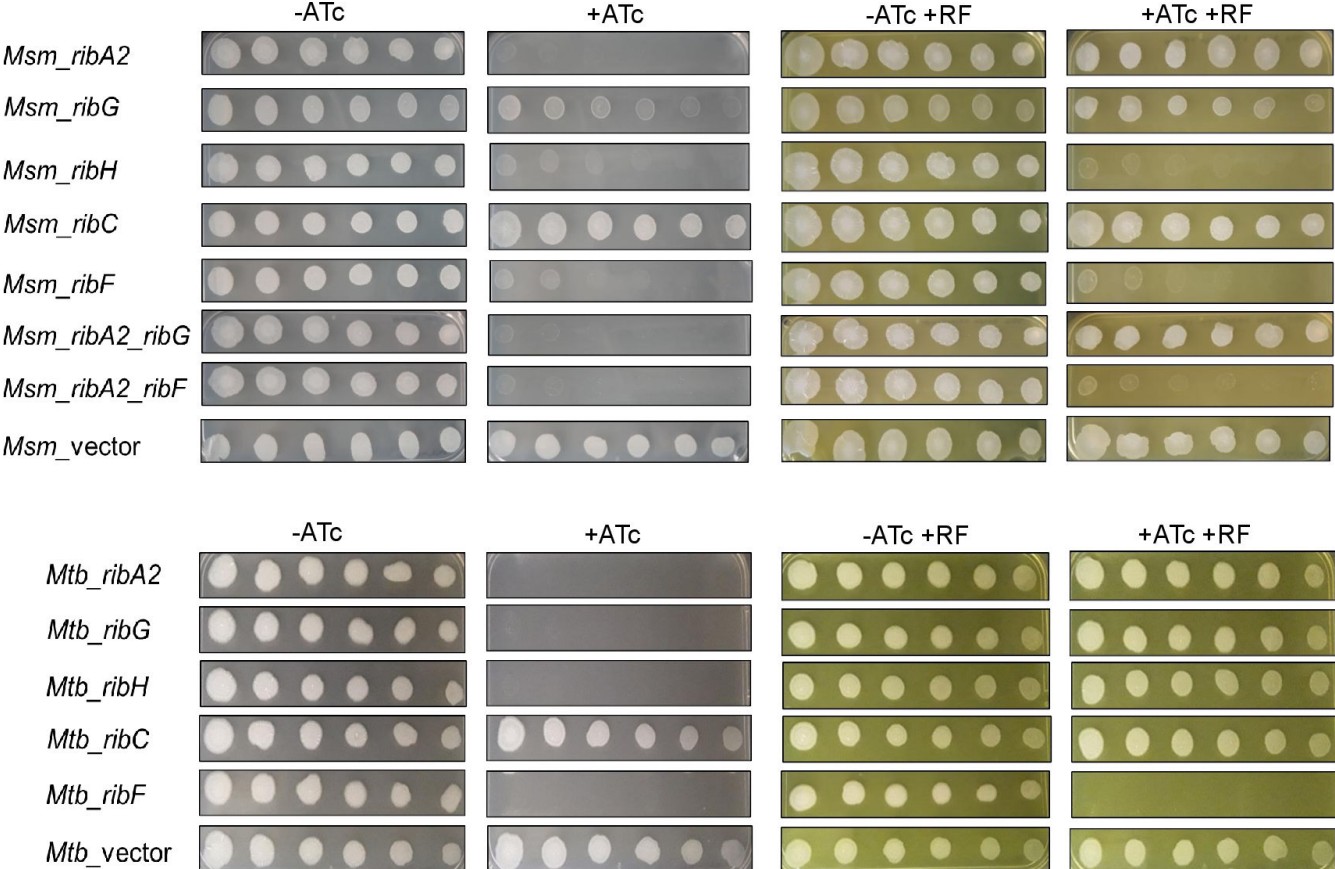

**FIG 2** Impact of RF pathway gene silencing on mycobacterial growth. Twofold serial dilutions of Msm (above) or Mtb hypomorphs (below) were spotted onto 7H10 agar in the absence (−ATc) or presence of ATc at 100 ng/mL (+ATc) to induce gene silencing. To assess the impact of exogenous RF supplement on growth, the hypomorphs were spotted, in parallel, on 7H10 agar containing RF at a concentration of 83 µM (Msm) or 21 µM (Mtb). The first spot in each row was seeded with ~5,000 bacilli.

CRISPRi vector system on the observed phenotypes, a Msm strain carrying a scrambled non-targeting sgRNA (*Msm*_NT) was assessed alongside the vector-only control (*Msm*_vector) and the RF pathway hypomorphs (Fig. S3). The lack of impact of ATc on growth of either Msm_NT or Msm_vector strain substantiated the use of *Msm*_vector as an appropriate negative control for subsequent experiments. Whereas RF pathway gene silencing was universally bactericidal in Mtb, only *ribA2* silencing was bactericidal in Msm, suggesting a greater ability of Msm to withstand the physiological impact of RF pathway disruption than Mtb. The bactericidal effect of *ribA2* silencing in both organisms confirmed that RibA1 could not substitute for the GTP cyclohydrolase II function of RibA2 under the conditions tested.

## Impact of RF pathway silencing on transcript, protein, and RF levels

To correlate these phenotypes with the expression of the target genes and their encoded proteins, the levels of *sigA*-normalized transcripts were measured in the Msm or Mtb hypomorphs in the presence or absence of ATc (Fig. 3 and 4). Analysis of expression levels of RF pathway genes in the *Msm*_NT and *Msm*_vector strains revealed an impact of the CRISPRi system on expression of *ribA2*, with both strains showing a ~2- to 3-fold increase in *ribA2* expression in the presence of ATc (Fig. S5). However, the expression of the other pathway genes was unaffected. Operonic structure is a potential complication in transcriptional silencing as polar effects on neighboring genes can confound the interpretation of phenotypes. For this reason, transcript levels in all Msm hypomorphs

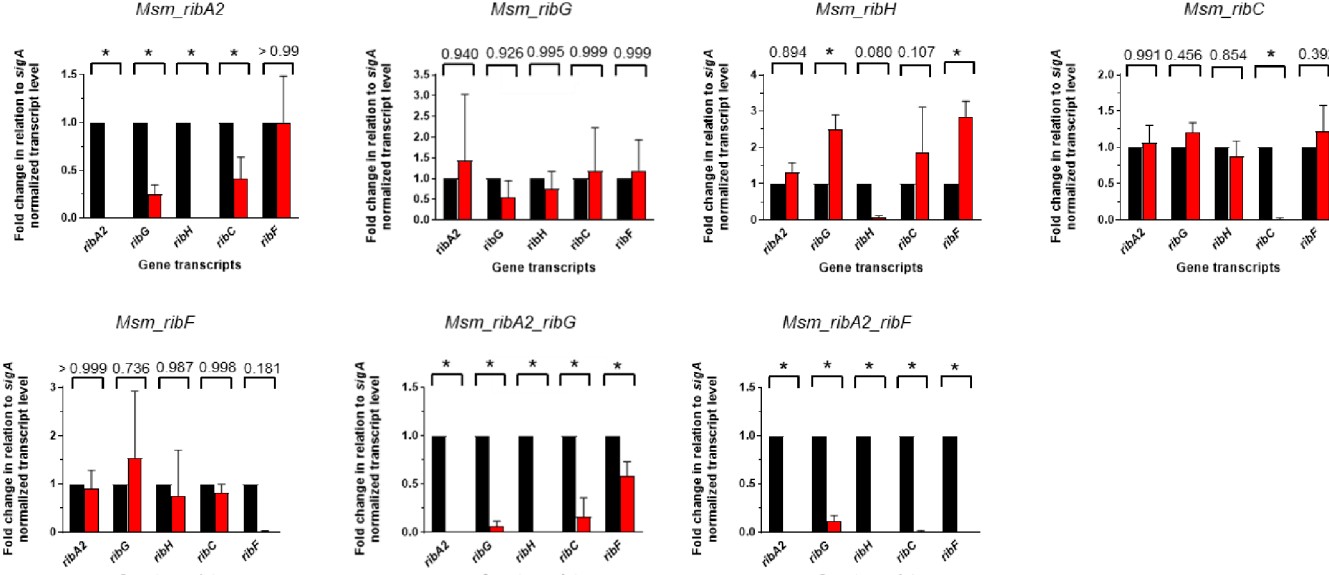

**FIG 3** Impact of RF pathway gene silencing on the expression of the target/s and other genes within the pathway in Msm showing fold gene expression in the presence (red bars) vs absence of ATc (black bars). Error bars represent the SD derived from three biological replicates. Statistical comparisons were performed using a two-way ANOVA and either the Sidak's or Dunnett's multiple comparison test whereby statistical significance is represented by $P < 0.005$, denoted by an asterisk.

were measured for the cognate gene/s as well as the other RF pathway genes comparing transcript levels with or without ATc (Fig. 3). Under the conditions tested—that is, using a starting inoculum at an $OD_{600}$ of 0.06 by diluting bacteria from a late log-phase culture and exposing the culture to ATc for 24 h—ATc-dependent knockdown of the target gene was observed for all five pathway genes in Msm (Fig. 3). Whereas knockdown of *ribG*, *ribC*, or *ribF* had no significant impact on the expression of other pathway genes, silencing of *ribA2* profoundly suppressed *ribH* expression while also knocking down expression of *ribG* and *ribC* without affecting *ribF* expression. Although knockdown of *ribH* upon silencing of *ribA2* likely reflects a polar effect, the reasons underlying the concomitant knockdown of *ribG* and *ribC* are unclear. Silencing *ribH* also influenced the expression of other pathway genes, resulting in upregulation of *ribG* and *ribF*. The levels of pathway transcripts in the dual hypomorphs reflected the additive effects of knockdown of the individual genes with *ribA2*, *ribG*, *ribH*, and *ribC* expression repressed by ≥80% in both strains and >99% knockdown of *ribF* in the *Msm_ribA2-ribF* strain. The levels of *ribH*, *ribG*, and *ribC* transcript were also reduced by >99% in this strain.

The impact of knockdown of individual RF pathway genes on the expression of pathway genes was then assessed in Mtb (Fig. 4). Consistent with the operonic arrangement, transcriptional silencing of *ribA2* led to knockdown of *ribA2* as well as *ribH*, but unlike Msm, *ribA2* silencing in Mtb did not affect the expression of other genes in the pathway. Transcriptional silencing of *ribA2*, *ribH*, or *ribC* was specific, showing highly significant knockdown of the cognate gene. Although ATc-dependent knockdown of *ribG* or *ribF* did not reach statistical significance in their corresponding hypomorphs under the conditions tested, the survival of Mtb_*ribG* and Mtb_*ribF* was, nonetheless, impaired under the conditions used to assess growth and survival (Fig. S4). Silencing of *ribF* also led to upregulation of both *ribG* and *ribH*.

Since other factors such as the abundance and stability of the encoded protein can also impact on the phenotypic outcome of transcriptional silencing of a target gene, we used targeted proteomics to measure the levels of the RF pathway proteins in the Msm hypomorphs. Production of sufficient biomass for proteomic analysis required exposure of a higher density culture to ATc than that used for growth/survival phenotyping (Fig. S3 and S4) or qRT-PCR analysis (Fig. 3 and 4) in liquid culture. We, therefore, investigated

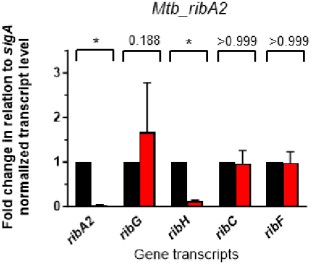

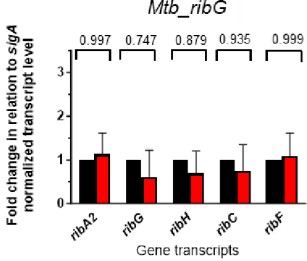

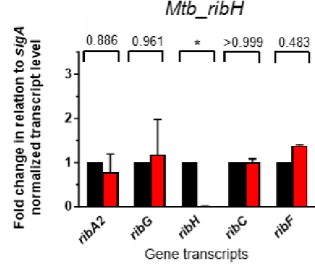

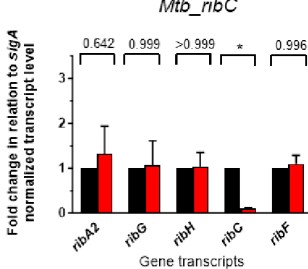

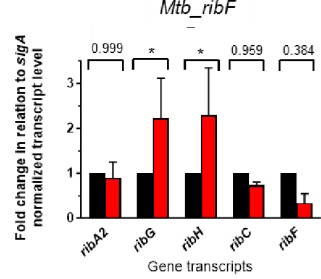

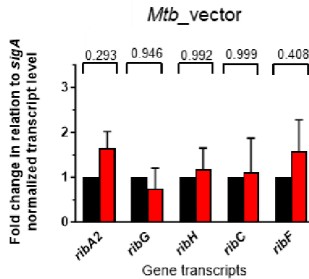

**FIG 4** Impact of RF pathway gene silencing on the expression of the target/s and other genes within the pathway in Mtb showing fold gene expression in the presence (red bars) vs absence of ATc (black bars). Error bars represent the SD derived from three biological replicates. Statistical comparisons were performed as described in the legend to Fig. 4, with statistical significance, represented by $P < 0.05$, denoted by an asterisk.

the inoculum effect on functionality of the CRISPRi system by monitoring the ATc responsiveness of the hypomorphs in cultures seeded with increasing numbers of bacilli (Fig. S6). For those Msm hypomorphs that showed ATc-dependent growth inhibition under standard conditions (i.e., when seeded at low cell density with bacteria from a late log-phase culture, Fig. S3), the phenotype was progressively dampened, albeit still evident in cultures seeded at higher cell densities. This suggested that target gene silencing would still occur under the culture conditions used for proteomic analysis, a conclusion consistent with the ATc-responsiveness of transcript levels of *ribA2* in Msm_*ribA2* when cultures were seeded at varying cell densities (Fig. S7). Accordingly, significant depletion of the RibA2 and RibC proteins was, indeed, observed in the *ribA2* and *ribC* hypomorphs, respectively, after 24 h silencing (Fig. 5A). The lack of an effect of *ribC* silencing on Msm growth (Fig. S3) under conditions resulting in significant depletion of RibC protein (Fig. 5A) is consistent with the relative invulnerability of RibC in this organism (Table S1). In contrast, a $3\log_{10}$ decline in CFU was observed after silencing of *ribA2* for 3 h.

RibG and RibH also trended toward lower levels in the corresponding hypomorphs treated with ATc, but, in these cases, the differences were not statistically significant. However, silencing of *ribF* was not associated with any change in the abundance of RibF protein. The dual hypomorphs showed a similar pattern: RibA2 and RibG levels were significantly reduced in *Msm-ribA2-ribG* and, while RibH also trended towards depletion, the effect did not reach statistical significance (Fig. 5B). However, in *Msm_ribA2_ribF*, RibA2 and RibH were both significantly depleted. Interestingly, as observed in the *ribF* hypomorph, the RibF protein was not significantly depleted in the dual *Msm_ribA2_ribF* hypomorph.

Targeted metabolomic analysis of whole-cell lysate (WCL) prepared under the same silencing conditions as used for the proteomics was carried out to ascertain the response of RF abundance to ATc-mediated knockdown of RF pathway genes (Fig. 6). This analysis revealed that RF levels were significantly reduced in the *Msm_ribA2*, *Msm_ribH*, *Msm_ribA2_ribG*, and *Msm_ribA2_ribF* hypomorphs.

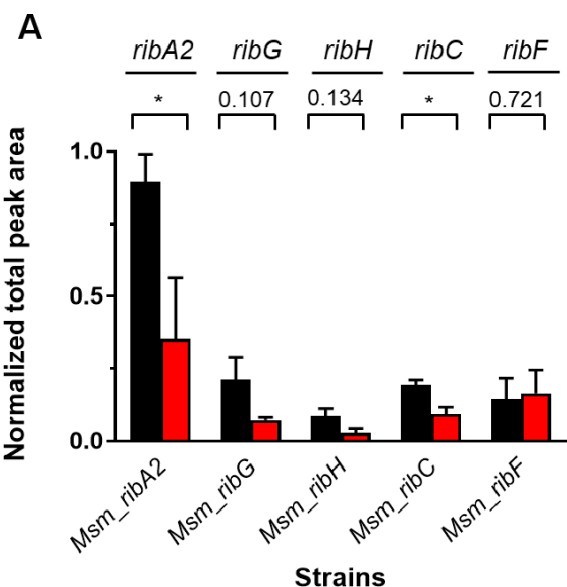
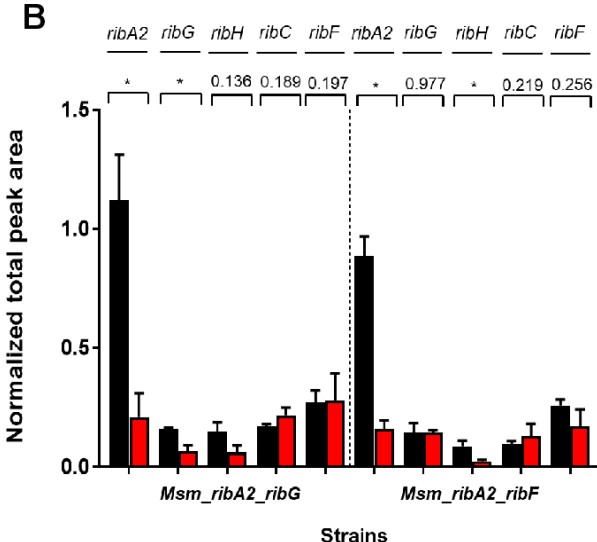

**FIG 5** Impact of RF pathway gene silencing on protein levels in Msm. Normalized total peak area representing relative protein abundance of RF pathway proteins in the absence (black) or presence of ATc (red) in single-gene (A) or duplexed (B) hypomorphs. Error bars represent the SD derived from three biological replicates. Statistical comparison was performed using paired Student's $t$-test whereby statistical significance is represented by $P < 0.05$, denoted by an asterisk.

## Variable rescue of hypomorphs from RF auxotrophy by exogenous flavins

Tn-seq analysis had revealed that the riboflavin biosynthesis genes of Mtb were not essential in MtbYM rich medium which contains RF supplement at a concentration of 53 µM (45). This suggested that Mtb can transport and assimilate exogenous RF even though it lacks a canonical transporter. This finding provided a means of potentially linking the phenotypes of the hypomorphs with silencing of the corresponding pathway gene/s. The RF dose-responsiveness of the hypomorphs and controls, grown in liquid culture at a fixed concentration of ATc (50 ng/mL) and over an RF concentration range of 1.3 µM – 2.7 mM, was determined using a MABA assay (Fig. S8 and S9). RF supplementation had no effect on the growth of the wild-type strains and, as expected, exogenous RF could not rescue Msm or Mtb from the growth inhibitory effect of *ribF* silencing. The minimum concentration of RF required to restore growth of the Msm *ribA2*, *ribG* and *ribA2_ribG* hypomorphs to ≥80% of the level of wildtype was 42–83 µM, whereas the Mtb *ribA2*, *ribG* and *ribH* hypomorphs required a lower concentration of RF for complete rescue (10–21 µM). In contrast, the Msm *ribH* hypomorph was only partly rescued at an RF concentration ≥1.3 mM.

We then assessed growth of the hypomorphs on solid media with RF supplementation (83 µM for Msm or 21 µM for Mtb), with or without ATc (Fig. 2). The results were broadly consistent with those in liquid culture: complete rescue of the Mtb *ribA2*, *ribG* and *ribH* hypomorphs as well as the Msm *ribA2*, *ribG,* and *ribA2_ribG* hypomorphs, but minimal rescue of Msm *ribH* and *ribA2_ribF* hypomorphs in the presence of ATc.

Unlike RF, the coenzymes FMN and FAD were unable to rescue any of the hypomorphs from the growth inhibitory effects of RF pathway disruption observed on solid or in liquid culture when supplied exogenously at concentrations of 44 µM or 25 µM, respectively. Furthermore, no rescue of the Mtb *ribF* hypomorph by FMN or FAD was observed at coenzyme concentrations up to 1.3 mM (FAD) or 2.2 mM (FAD) (data not shown). Thus, exogenous RF can be transported and assimilated by mycobacteria, but FMN and FAD cannot.

We then investigated the effect of exogenous RF on the transcript levels of RF pathway genes and on the levels of the corresponding proteins in Msm; however, this was found to have no significant impact on either transcript or protein levels (Fig. S10).

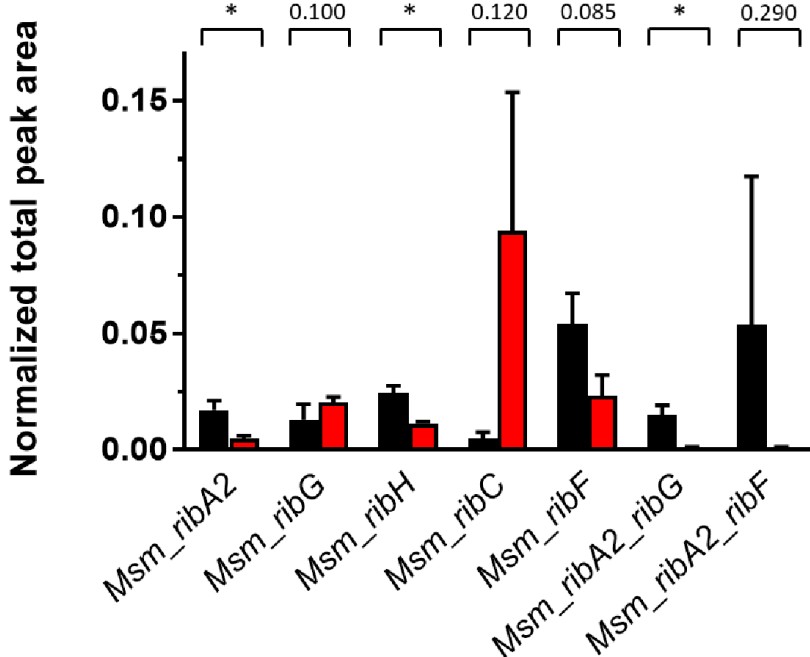

**FIG 6** Normalized total peak area representing relative abundance of RF in Msm strains prior to (black) or after ATc-mediated gene silencing (red) in single-gene and dual hypomorphs. Error bars represent SD derived from three biological replicates. Statistical comparison was performed using paired Student's *t*-test whereby statistical significance is represented by *P* < 0.05, denoted by an asterisk.

## Redundancy of lumazine synthase activity in Msm

We hypothesized that the inability of exogenous RF to rescue the ATc-dependent growth phenotype of the Msm *ribH* hypomorph could be due to an off-target effect. Consistent with this hypothesis, we were able to delete Msm *ribH* in the absence of RF supplement, confirming the dispensability of this gene, and implicating *MSMEG_6598*, which shares 39% sequence identity with *ribH*, as an alternate lumazine synthase in Msm. To test this, we compared the effect of silencing *MSMEG_6598* in wild-type Msm vs the Δ*ribH* mutant and showed that, while *MSMEG_6598* silencing had no effect on growth of wild-type Msm, it was growth inhibitory in a Δ*ribH* background. RF supplement restored growth of the *Msm_ΔribH_MSMEG_6598* hypomorph in the presence of ATc and enhanced growth of the parental Δ*ribH* strain, thus confirming redundancy in lumazine synthase function in Msm (Fig. 7). Furthermore, we verified that in *Msm_ribH*, the sgRNA knocked down the expression of *ribH* while the expression of the second lumazine synthase-encoding gene, MSMEG_6598, was unaffected (Fig. S11).

## DISCUSSION

In this study, we used a transcriptional silencing approach to investigate RF biosynthesis and utilization in Mtb and Msm. We uncovered both similarities and differences between the RF pathways in Mtb and Msm. RibA2 were shown to be essential in both organisms, raising questions about whether and under which conditions the second putative GTP cyclohydrolase II, RibA1, contributes to the biosynthesis of RF. In contrast, RibH activity is provided by two paralogs in Msm, whereas Mtb has only one essential RibH enzyme.

RF pathway disruption by CRISPRi was shown to be bactericidal in Mtb with silencing of *ribA2*, *ribG*, or *ribH* resulting in 2–3.5log$_{10}$ reduction in CFU over 4 days. Msm appeared less vulnerable to RF pathway disruption with only *ribA2* silencing having a bactericidal effect under the conditions tested. In Msm, however, silencing of *ribA2* profoundly affected the expression of other genes in the RF biosynthesis pathway, resulting in

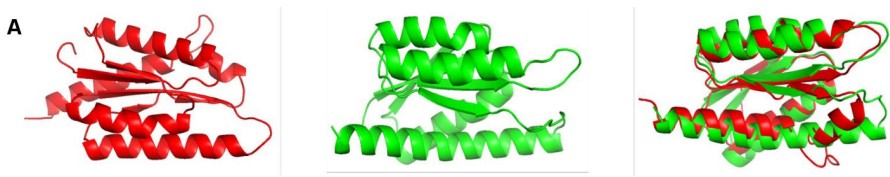

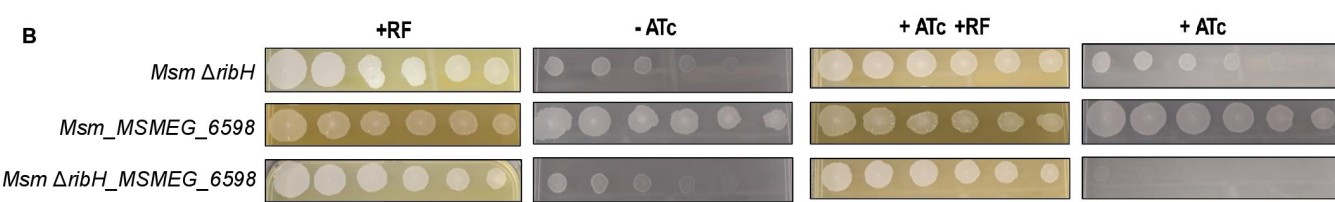

FIG 7  Evidence for redundant RibH function in Msm. (A) Predicted tertiary protein structure of RibH (*MSMEG_3073*, red), *MSMEG_6598* (green) individually (left and center, respectively) and superimposed (right), determined using Phyre2. (B) Growth phenotyping of Msm hypomorphs in *MSMEG_6598* in wildtype vs Δ*ribH* backgrounds. Twofold serial dilutions of strains were spotted on 7H10 media in the absence (−ATc) or presence of ATc (+ATc, 100 ng/mL), with or without exogenous RF (83 µM).

downregulation of *ribH*, *ribG,* and *ribC* expression. While the knockdown of *ribH* can be explained by a polar effect of *ribA2* silencing, the regulatory mechanism/s governing how *ribG* and *ribC* expression is influenced by transcription of *ribA2* is unclear. Our results nonetheless suggest that the vulnerability of RibA2 in Msm inferred from CRISPRi-mediated silencing is exaggerated by polar and other effects on the expression of other genes in the RF biosynthesis pathway. In contrast to Msm, knockdown of *ribA2* in Mtb only impacted the expression of *ribH* through a polar effect but did not affect the expression of other pathway genes. The vulnerability of RibA2 in Mtb is therefore likely exaggerated only by the polar effect on *ribH*. These results point to differences in the regulatory mechanisms governing the expression of RF pathway genes in the two mycobacterial species.

Msm has been classified as an RF "overproducer" (46) the term applied to organisms that accumulate RF at a concentration >10 mg/L (47). Like other coenzymes, FMN and FAD have been shown to be remarkably stable in *E. coli* and other microbes; this feature allows biosynthesis to occur at the minimal rate needed to compensate for their dilution by growth as cells divide (48). The authors of that study also showed that coenzyme pools are maintained at levels higher than required for growth. Assuming the flavin coenzyme longevity established by Hartl et al. (48) in other organisms is conserved in mycobacteria, then the availability of a reservoir of RF precursor to replenish the pools of FMN and FAD could potentially buffer mycobacteria against the lethal effects of pathway disruption. However, this requires the existence of a mechanism/s to sense and respond to coenzyme depletion. Given their lack of recognizable FMN riboswitches, whether and how this might occur in mycobacteria remains an open question.

The regulation of RF biosynthesis has been elucidated in some bacteria, where positive and negative regulatory mechanisms operating at the levels of modulation of enzyme activity, transcription, and/or translation have been described (3, 46). The rate-limiting step in RF biosynthesis in *B. subtilis* is GTP cyclohydrolase II, but very little is known about potential feedback control in this or other organisms (46), so it is unclear whether this mechanism applies in mycobacteria. The FMN riboswitch plays a key role in regulating polycistronic or monocistronic *rib* genes and RF transporters in some bacteria, enabling RF biosynthesis and transporter gene expression to be coupled to intracellular levels of FMN (43, 46, 49, 50). When supplied at a level sufficient to rescue Msm from RF auxotrophy (83 µM), exogenous RF had no impact on transcription of any of the pathway genes, or on abundance of the encoded proteins, except for RibC, which was slightly depleted.

An association between iron levels and expression of riboflavin supply pathways has been established in some bacteria, including *H. pylori*, *Clostridium acetobutylicum,* and *Caulobacter crescentus,* with iron deficiency leading to increased flavin synthesis (3, 46). In *C. acetobutylicum*, the *rib* operon is also positively regulated by the carbon starvation regulator, CsrA (51). Other regulatory mechanisms reported include increased expression and RF secretion by *sigH* overexpression in the actinomycete, *C. glutamicum* (52). Whether mycobacteria engage analogous mechanisms remains an open question.

The attractiveness of RF biosynthesis enzymes as TB drug targets depends on various factors, including whether Mtb can access RF from infected host tissue. Although Mtb appears to lack homologs of RF transporters found in other bacteria (3, 43, 53, 54), a recent study suggested that Mtb can import and assimilate RF as evidenced by the apparent ability of RF supplement to rescue the growth of transposon mutants in RF pathway genes *in vitro* (45). The lack of canonical transporters notwithstanding, we confirmed the ability of Mtb and Msm to import RF from the culture medium and established the minimum levels required to rescue these mycobacteria from RF auxotrophy (20–80 µM). The level for the rescue of Mtb (20 µM) far exceeds the baseline level of RF in blood plasma of humans (11–33 nM) (55) suggesting that subversion of the essentiality of riboflavin biosynthesis in Mtb by scavenge of RF from the host would be highly unlikely. This conclusion is consistent with the profound fitness defects of transposon mutants of Mtb in *ribA2* and *ribG* reported recently in mice (56). Transcriptional silencing of *ribF* was also strongly bactericidal in Mtb *in vitro*, resulting in a $3.5\log_{10}$ decline in CFU within 3 days. However, FMN and FAD were unable to rescue Msm or Mtb from the inhibitory effects of *ribF* silencing suggesting that, unlike the RF auxotroph, *Listeria monocytogenes*, which uses the RibU transporter to scavenge FMN and FAD from the host cell cytoplasm to grow intracellularly (54), mycobacteria cannot import these coenzymes. Together, these results provide *in vitro* genetic validation for targeting RF biosynthesis and utilization for TB drug discovery.

During the course of this work, Dey et al. reported the results of a study in which they adopted a complementary approach to show that overexpressing *ribA2*, *ribG*, *ribH*, or *ribF* in Mtb or Msm resulted in variable potentiation MAIT cell activation *in vitro* and overexpression of *ribA2* led to attenuation of Mtb virulence *in vivo* (57). In their study, unspecified RF metabolite overexpression was inferred from spectral analysis of the supernatant of Msm cultures overexpressing *ribA2*, but the level of RF was not measured in any of their overexpressing strains (57). In our work, we used targeted metabolomics to measure the impact of silencing each step of the pathway on the level of RF in Msm and demonstrated significant depletion of RF in the *ribA2*, *ribA2_ribG*, and *ribH* hypomorphs under the conditions tested. However, the full extent of metabolomic derangement resulting from RF pathway disruption in mycobacteria has yet to be established. This issue is central to the question of optimal target selection in terms of risk-benefit given the potential trade-off between the efficacy of Mtb kill and MAIT cell activation vs antagonism. For example, does inhibition of RibA2 perturb the levels of GTP and, if so, what are the knock-on consequences on other GTP-dependent cellular functions such as folate metabolism and the stringent response? Likewise, how does RibA2 inhibition affect the levels of ribulose 5-phosphate, and hence, pentose phosphate pathway function? Furthermore, how responsive are levels of 5-AR-U to pathway disruption, and does this impact on MAIT cell activator vs antagonist production and/or deazaflavin levels and, in turn, susceptibility of Mtb to nitroimidazole drugs? Likewise, how do the levels of DMRL and its derivatives respond to pathway disruption? The hypomorphs and analytical methods described here provide a useful resource for addressing these questions.

## MATERIALS AND METHODS

### Media and culture conditions

All strains generated in this study were derivatives of *E. coli* DH5α, Msm mc$^2$155 or Mtb H37RvMA (Table S6). *E. coli* DH5α was grown in Luria-Bertani (LB) medium containing 10 g/L tryptone, 5 g/L yeast extract, and 10 g/L NaCl and plated onto Luria-Bertani agar (LB agar) containing LB media components supplemented with 15 g/L agar. Mtb and Msm strains were cultured in Middlebrook 7H9 medium supplemented with 100 mL oleic acid-albumin-dextrose-catalase (OADC) enrichment, 2 g/L glycerol, and 2.5 mL 25% Tween 80, defined as 7H9 OADC. Cells were plated onto Middlebrook 7H10 agar plates supplemented with 100 mL OADC and 5 mL glycerol. Msm strains were grown in Erlenmeyer flasks in a rotating incubator set at 100 rpm at 37°C. Unless stated otherwise, Mtb strains were cultured in sealed cell culture flasks with no agitation at 37°C. Where appropriate, LB agar was supplemented with 50 µg/mL kanamycin (Km) and 7H10 was supplemented with 25 µg/mL Km. The Msm Δ*ribH* mutant was supplemented with 50 µg/mL hygromycin (Hyg) and the Msm Δ*ribH MSMEG_6598* hypomorph with 50 µg/mL Hyg and 25 µg/mL Km. To induce gene silencing in Msm or Mtb hypomorphs, ATc was added at 100 ng/mL and approximately $10^4$–$10^5$ cells were used as the starting inoculum, unless indicated otherwise. All chemicals were purchased from Merck.

### Construction of hypomorphs and gene knockout strain

The oligonucleotides, plasmids, and strains used in this study are described in Tables S2 to S7. The ATc-inducible (P$_{tet}$) *Streptococcus thermophilius* (Sth) dCas9 and P$_{tet}$ sgRNA system for inducible CRISPRi developed by Rock et al. (35) was used to generate hypomorphs in Msm and Mtb using a set of relevant sgRNAs shown in Fig. S2 and as previous described (58). Briefly, two oligonucleotides complementary to the target gene of interest in Msm (Table S2) or Mtb (Table S3) were synthesized, annealed, and cloned into either pLJR962 or pLJR965 (35). The vectors were transformed into *E. coli* and extracted to confirm the presence of the sgRNA by Sanger sequencing using the primer, 5′-TTCCTGTGAAGAGCCATTGATAATG-3′. The resulting construct was electroporated into the parental mycobacterial strain and selected on 25 µg/mL Km. Sets of 54 Msm and 47 Mtb hypomorphs, each carrying an individual sgRNA, were created. The parental strain and a derivative carrying the empty vector (without a targeting sgRNA) were used as negative controls, and a hypomorph targeting the *mmpL3* gene was used as a positive silencing control. For duplexing, the PCR-amplified sgRNA cassette containing *Sap*I restriction sites was digested and cloned into the sgRNA *Sap*I-digested construct backbone. The construct was sequenced to confirm the presence of both sgRNA cassettes. The ORBIT system (59) was used to delete *ribH* in Msm using a targeting oligonucleotide containing 70 bp of homology to *ribH* flanked by the BxbI phage *attP* sequence (Table S4). The oligonucleotide was co-electroporated with pKM464 into the Msm pKM461 background strain and selected on 7H10 plates containing 10% sucrose and 50 µg/mL Hyg.

### RNA extraction and expression analysis by qRT-PCR

Strains were inoculated from late log-phase cultures at $4 \times 10^5$–$7 \times 10^5$ CFU/mL in 20 mL 7H9 OADC containing either 100 ng/mL ATc or no ATc and supplemented with Km where necessary. Cultures were incubated for 24 h for Msm or 4 days for Mtb. Cells were harvested, washed, and resuspended in TRI reagent. The cells were transferred to homogenization microtubes containing 0.1 mm silica lysing matrix (Biospec) and lysed using the FastPrep FP120 BIO 101 Savant set at a speed of 5 for 30 s. This was repeated three times with a 2 min incubation on ice between each cycle. Samples were further processed according to the manufacturer's instructions using the Direct-zol RNA extraction kit (Inqaba Biotechnical Industries (Pty) Ltd, South Africa).

Total RNA was treated with TURBO DNAse (Invitrogen) to remove contaminating DNA and 250 ng of the RNA was converted to cDNA using the SuperScript IV Reverse

Transcriptase (Thermo Fischer Scientific). Power SYBR Green PCR master mix (Thermo Fischer Scientific) was used to amplify the target region of interest using the primer pairs described in Table S7 and quantified on the PikoReal real-time PCR system (Thermo Fischer Scientific). A no-reverse-transcriptase control was included to verify the absence of DNA contamination. Transcript levels of the target gene were normalized to *sigA* for both Msm and Mtb and analyzed as the log fold change by calculating $\Delta\Delta Ct$. For each strain, the fold gene expression in the presence of ATc was calculated relative to the no-ATc control.

To assess operonic structure, RNA was extracted from wild-type Msm or Mtb as described above. PCR was conducted using primers to amplify the junctions between adjacent genes (Table S8) to determine if they constitute a single transcriptional unit. Genomic DNA, for use as a control, was prepared by the cetyltrimethylammonium method, as previously described (60).

## Phenotypic assessment of RF pathway gene silencing

For spotting assays, cultures were grown to an $OD_{600} = 0.5–0.6$ ($\sim 1 \times 10^7$ CFU/mL). A twofold serial dilution was performed, after which 5 µL of each dilution was spotted onto 7H10 agar (containing Km, where appropriate) supplemented with either no ATc or 100 ng/mL ATc. Msm plates were incubated for 2–3 days, whereas Mtb plates were incubated for ~9 days at 37°C before imaging. To establish the minimum ATc concentration required to inhibit growth an MABA assay was employed: ATc (400 ng/mL) in 7H9 OADC was diluted 2-fold in a clear well round bottom 96-well microtitre plate. Mtb or Msm was grown to an $OD_{600}$ ~0.5 and diluted 1,000-fold before adding an equal volume to each well, totaling 100 µL. The plate was sealed and incubated at 37°C for 24 h for Msm and 6 days for Mtb. Thereafter, 10 µL of 0.01% Alamar blue was added and incubated for a further 9 h for Msm or 12 h for Mtb. Fluorescence was recorded using the SpectraMax i3x (Molecular Devices) (excitation 540 nm; emission 590 nm) and the percentage survival was calculated using the equation: (fluorescence$_{well}$/average fluorescence of vector control) $\times 100$.

To assess the inoculum effect (i.e., the impact of cell density) on the efficacy of the inducible CRISPRi system in targeted gene silencing, strains were grown to late log-phase ($OD_{600}$ ~0.5; $1 \times 10^7$ CFU/mL) and inoculated into 7H9 OADC in the presence of 100 ng/mL ATc or without ATc so that the $OD_{600}$ values of the starting inocula were 0.004, 0.016, 0.06, or 0.25, corresponding to $\sim 5 \times 10^4$, $\sim 3 \times 10^5$, $\sim 7 \times 10^5$, and $\sim 1 \times 10^6$ CFU/mL. For Msm cultures, the $OD_{600}$ was measured every 3 h between 12 and 24 h, and for Mtb, $OD_{600}$ measurements were taken every 24 h over 7 days.

## ATc responsiveness of growth and survival of hypomorphs

To monitor the impact of gene silencing on the growth and survival of the hypomorphs in liquid culture, a starting inoculum of $5 \times 10^4$ cells ($OD_{600} = 0.004$) was inoculated from a late log-phase culture ($OD_{600} = 0.5$) into 50 mL 7H9 OADC. Each strain was either treated with or without ATc (at 100 ng/mL) and cells were harvested every 3 h over 24 h (Msm) or every 24 h over 7 days (Mtb). At each time point, the $OD_{600}$ was recorded, and cells were washed three times before performing a dilution series and plating for CFU enumeration. CFUs were enumerated after incubation for 5–7 days (Msm) or 3–4 weeks (Mtb).

## Growth of Msm cultures for targeted proteomics and metabolomics

Glycerol stocks of Msm hypomorphs were spread onto 7H11 agar plates containing 25 µg/mL Km. The plates were incubated for 4–5 days at 37°C, scraped and resuspended into 400 mL of LB media with Km, and grown for a further 24 h at 37°C in a roller bottle. Cells were harvested and washed with sterile water to remove media. Half the cells were inoculated into roller bottles containing fresh media supplemented with ATc (100 ng/mL), and the other half was retained as an uninduced control. Induced cultures

were grown for an additional 24 h and then harvested. Cells were harvested by allowing them to settle and then transferred to a conical tube. Cells were washed three times with water, and the cell pellet was stored at −80°C until further processing. The wild-type strain was grown and treated in the same way with no antibiotic supplement. A separate culture of the wild-type strain with RF added was performed to determine the effect of RF in the protein levels of the RF enzymes. All cultures were grown in triplicate.

## Preparation of WCL for proteomic and metabolomic analyses

Cell pellets were lysed by probe sonication (12 cycles of 60 s ON, 90 s OFF, 50% duty cycle) in 5 mM ammonium acetate supplemented with 0.6 µg/mL RNaseA, 0.6 µg/mL DNAse I, and Complete EDTA-free protease inhibitors (Roche). After sonication, cell debris was removed by centrifugation at 3,000 rpm. WCLs were passed through a 3 kDa MWCO Amicon ultrafiltration device and washed three times with 10 mM ammonium acetate. The flow through (<3 kDa fraction) was lyophilized and stored at −20°C for targeted metabolomic analysis. The retentate (WCL protein fraction) was quantified using the Bicinchoninic acid assay (BCA) (Pierce 23227) and stored at −20°C for targeted proteomics.

## Targeted proteomic analysis by multiple reaction monitoring-mass spectrometry

Method development: Thirty micrograms of WCL protein fraction was subjected to in-gel trypsin digestion as described elsewhere (61). Digested samples were resuspended in 30 µL of loading buffer (5% acetonitrile, 0.1% formic acid in water). Method development for multiple reaction monitoring-mass spectrometry (MRM-MS) was performed using an empirical approach and a pooled sample of all digested samples. Briefly, a method containing all suitable tryptic peptides (6–20 aa long) and at least 5 transitions (y-ions) per peptide for each protein target was created in Skyline (62). Peptides containing cysteines were not included. The method was exported to MassLynx (Waters corp) and trialed in a Xevo TQ-S mass spectrometer (Waters corp.) coupled to a M-class UPLC (Waters corp.) and an ionKey source. Peptides were separated in an ion key column (iKey, 150 µm × 50 mm Peptide BEH C18, 13 Å, 1.7 µm) using a 10 min linear gradient from 5% Buffer B (0.1% formic acid in acetonitrile), 95% Buffer A (0.1% formic acid in water) to 45% buffer B, 55% Buffer A. Following the gradient, the ionKey was washed at 95% Buffer B, 5% Buffer A for 5 min followed by equilibration at 5% Buffer B, and 95% Buffer A for 5 min. The total time for each injection was 20 min. Data were collected with a minimum of 17 points per peak. Raw data were exported into Skyline, and the peptides with the highest intensity peaks were selected for further optimization. Optimization included trials of alternate transitions and identification of optimal collision energy. After the trials, at least two top peptides per target protein were selected for the final method. For the final method, synthetic peptides containing a heavy labeled C-terminal lysine (K) or arginine (R) were purchased (New England Peptides) and used as internal standards for each selected peptide. Limit of detection (LOD) and Limit of quantification (LOQ) were calculated using a linear regression method as described previously (63). See Table S9 for information on the peptides used in the final method.

## Relative quantification of RF pathway proteins by MRM-MS

A mix of the heavy labeled peptides containing 10–50 nM of each peptide was used to resuspend each of the digested samples at a final concentration of 1 µg/µL of digested protein. After resuspension, sample peptide concentration was estimated in a Nanodrop 1000c and the concentration was adjusted to ensure equal loading of all samples. The developed MRM method was used to obtain the normalized Total Peak Area (nTPA) for each of the monitored peptides. Briefly, raw data from each injection was imported into Skyline (62), and peak boundaries for each peptide were manually validated and adjusted if necessary. The normalized Total Peak Area (nTPA) of each peptide as calculated by Skyline was exported into Prism GraphPad.

## Targeted metabolomics by MRM-MS

An MRM-MS method for RF was developed using the precursor [$m/z$ 377.373 (M + H)] and transition ions [$m/z$ 172.2073 (M + H) and 198.1073 (M + H)] as described by Lock et al. (64). All other details were identical to those of the targeted proteomic analysis. Dry weight of the <3 kDa WCL fractions was obtained, and 30 µg of material from each of the strains and replicates was resuspended in 30 µL of loading buffer (5% acetonitrile, 0.1% formic acid in water) containing heavy labeled RF (dioxopyrimidine-13C4,15N2, Sigma) as an internal standard at a concentration of 1 ng/µL. The nTPA of the precursor ion was used for all quantitation purposes.

## Statistical analyses

Statistical analyses were performed by two-way ANOVA and either Sidak's or Dunnett's multiple comparison test. Statistically significant difference was set at either $P < 0.0001$ or $P < 0.05$, denoted by an asterisk. Calculations were performed using GraphPad Prism 7.04 statistical software (GraphPad Software, Inc., San Diego, CA). When assessing CFUs over time, the $P$ value is only shown for the end time point of either 24 h for Msm or 7 days for Mtb. nTPA of each peptide for the peptide MRM-MS, or nTPA of the RF MRM-MS for the Msm hypomorph strains were compared before and after induction of the hypomorph phenotype by Student's paired $t$-test. All analyses for these were performed using GraphPad Prism 9 with a statistical significance established at $P < 0.05$.

## ACKNOWLEDGMENTS

We thank Terry Kipkorir, Helena Boshoff, and David Sherman for helpful discussions and Jeremy Rock for kindly providing pLJR965 and pLJR962.

This research was supported by an Oppenheimer Fellowship from the Oppenheimer Memorial Trust (to V.M.), and grants from the Bill and Melinda Gates Foundation (INV-004757), the South African Medical Research Council (to V.M.), the Department of Science and Innovation and National Research Foundation of South Africa (to V.M.), the University of Cape Town (to M.D.C.), NIAID (R01AI147954, to D.M.L.), a subaward from the University of Chicago (Erin Adams, PI, R01AI147954) (AWD100279, to K.M.D. and C.M.) and internal funds from Colorado State University (to K.M.D.).

## AUTHOR AFFILIATIONS

[1]Molecular Mycobacteriology Research Unit, Institute of Infectious Disease and Molecular Medicine & Department of Pathology, University of Cape Town, Cape Town, South Africa
[2]Department of Microbiology, Immunology and Pathology, Colorado State University, Fort Collins, Colorado, USA
[3]Wellcome Centre for Infectious Disease Research in Africa, University of Cape Town, Cape Town, South Africa
[4]Oregon Health and Science University, Portland, Oregon, USA
[5]Portland VA Medical Center, Portland, Oregon, USA

## AUTHOR ORCIDs

Melissa D. Chengalroyen  http://orcid.org/0000-0002-3712-3587
Digby F. Warner  http://orcid.org/0000-0002-4146-0930
Karen M. Dobos  http://orcid.org/0000-0001-7115-8524
Valerie Mizrahi  http://orcid.org/0000-0003-4824-9115

## FUNDING

| Funder | Grant(s) | Author(s) |
|---|---|---|
| Ernest Oppenheimer Memorial Trust (Oppenheimer Memorial Trust) | Harry Oppenheimer Fellowship, Oppenheimer Fellowship | Valerie Mizrahi |
| Bill and Melinda Gates Foundation (GF) | INV-004757 | Valerie Mizrahi |
| South African Medical Research Council (SAMRC) | Extramural Unit | Valerie Mizrahi |
| National Research Foundation (NRF) | DSI/NRF Centre of Excellence for Biomedical TB Research | Valerie Mizrahi |
| Department of Science and Innovation, South Africa (DSI) | DSI/NRF Centre of Excellence for Biomedical TB Research | Valerie Mizrahi |
| HHS \| National Institutes of Health (NIH) | R01AI147954 | Karen M. Dobos |
| University of Cape Town (UCT) | Emerging Researcher | Melissa D. Chengal-royen |
| Colorado State University (CSU) | Internal funds | Karen M. Dobos |
| HHS \| NIH \| National Institute of Allergy and Infectious Diseases (NIAID) | R01AI147954 | David M. Lewinsohn |

## AUTHOR CONTRIBUTIONS

Melissa D. Chengalroyen, Conceptualization, Data curation, Funding acquisition, Investigation, Methodology, Validation, Visualization, Writing – original draft, Writing – review and editing | Carolina Mehaffy, Conceptualization, Formal analysis, Funding acquisition, Investigation, Methodology, Visualization, Writing – review and editing | Megan Lucas, Conceptualization, Funding acquisition, Project administration, Resources, Supervision, Writing – review and editing | Niel Bauer, Investigation | Mabule L. Raphela, Investigation | Nurudeen Oketade, Investigation, Supervision | Digby F. Warner, Conceptualization, Supervision, Writing – review and editing | Deborah A. Lewinsohn, Writing – review and editing | David M. Lewinsohn, Conceptualization, Funding acquisition, Writing – review and editing | Karen M. Dobos, Conceptualization, Formal analysis, Funding acquisition, Project administration, Resources, Writing – review and editing | Valerie Mizrahi, Conceptualization, Funding acquisition, Project administration, Resources, Supervision, Writing – original draft

## ADDITIONAL FILES

The following material is available online.

### Supplemental Material

**Supplemental material (Spectrum03207-23-s0001.pdf).** Fig. S1 to S11; Tables S1 to S9.

### Open Peer Review

**PEER REVIEW HISTORY (review-history.pdf).** An accounting of the reviewer comments and feedback.

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
