## [Reviewer comments · Microbiology Spectrum]

Microbiology Spectrum

Modulation of riboflavin biosynthesis and utilization in mycobacteria

Melissa Chengalroyen, Carolina Mehaffy, Megan Lucas, Neil Bauer, Mabule Raphela, Nurudeen Oketade, Digby Warner, Deborah Lewinsohn, David Lewinsohn, Karen Dobos, and Valerie Mizrahi

Corresponding Author(s): Valerie Mizrahi, University of Cape Town

Review Timeline:

Submission Date:	September 6, 2023
Editorial Decision:	October 25, 2023
Revision Received:	May 12, 2024
Accepted:	May 17, 2024

Editor: Gyanu Lamichhane

Reviewer(s): The reviewers have opted to remain anonymous.

Transaction Report:

DOI: <https://doi.org/10.1128/spectrum.03207-23>

Re: Spectrum03207-23 (Modulation of riboflavin biosynthesis and utilization in mycobacteria)

Dear Dr. Valerie Mizrahi:

Thank you for the privilege of reviewing your work.

Two referees with expertise in mycobacteriology have reviewed your manuscript and provided helpful feedback. Below you will find all comments and instructions from the Spectrum editorial office.

Revision Guidelines

Sincerely,
Gyanu Lamichhane
Editor
Microbiology Spectrum

Reviewer #1 (Comments for the Author):

In this work, Chengalroyen et al perform a detailed CRISPRi analysis to study key enzymes in the riboflavin pathway. The authors compared the phenotypes of the two related Mycobacterium smegmatis and Mycobacterium tuberculosis. They find ribA2 or ribF to be bactericidal in Mtb, but only ribA2 is bactericidal in msmeg. Interestingly, exogenous riboflavin can rescue bactericidal effects of enzyme deletion, although the transporter is unknown.

This manuscript is timely, given that mycobacterium bacteria depend on RF for survival, the enzymes identified by the authors are promising anti-microbial drug targets. The manuscript is well-written and easy to understand. The authors do not over-state their results or significance of results.

The following concerns should be addressed:

1. The authors measure gene expression of RF mutants. However, there are substantial effects on growth of the different mutants - its hard to interpret the qPCRs if the bacteria are in different growth phases - can the authors perform experiment to measure RF gene expression at different growth phases, maybe this can also explain also the discrepancy of protein vs. mRNA of ribF? The effects of ribA2 on ribG and ribC? ribH on ribF? Also, can they add a non-specific target (as described by Rock et al: <https://www.nature.com/articles/nmicrobiol2016274>) for qPCR, and look at No-ATc and ATc+ to evaluate the general impact of the inserted CRISPR system on gene expression. From fig.S3 and S6 it does seem like the mutants are affected in ability to grow regardless to the presence of ATc.
2. Can the authors explain better some discrepancies between Fig 2 (growth of the 2-fold dilution colonies) and fig. S3 and S6 (growth curves in broth). Some of these seem to be important. For example, RibG shows partial growth deficiency between 2-fold dilutions. However, RibG in fig.S3 show not growth deficiency over 24h w/wo ATc, while RibG hypomorphs in fig S6 grow more poorly in all dilutions, w/wo ATc. In fig.2, RibH show a greater deficiency in growth, which does reflect in fig.S3 but not in the highest dilution of fig.S6, where the difference between w/wo ATc seems minor. Maybe perform the growth in broth also with or without RF (as in Fig. 2)?
3. Can the authors address the differences in growth of hypomorph RibH in Fig. 2 and deletion mutant Δ RibH in Fig. 6, especially in RF supplemented plates. Why does complete deletion of this gene, which is considered essential, allow growth in RF? Is it possible that the gRNA targets both ribH1 and ribH2 (msm6598)? Can qPCR be done to validate this?
4. There seems to be some interesting differences between mRNA, protein and metabolite levels (RF). Can the authors better explain how these fit with the RF synthesis pathway. For example, KD of RibF showed reduced mRNA (fig. 3) but not protein (fig. 4). However, if ribF is indeed KD it would be expected to result in increase of RF (fig. 5), as there should be no enzyme to convert RF.

Minor

1. Can the authors explain in more detail how were the guides for each gene chosen?
2. For Mtb - in Fig. S5 - will it be possible to look at how specific is the targeting on other genes in the pathways? For example, it will be interesting to see if ribA2 has also other off-targets like in msm?
3. In Fig 1, add location of RibA1 and RibH2 (MRMEG_5698) in msm, as they are referred to in the papers and fig. 6. Also, a step is missing to describe RibG bi-functionality - ARPP conversion to ArPP, which is generated to ArP by phosphatase.

Reviewer #2 (Comments for the Author):

This paper by Chengalroyen et al explores the potential for the exploitation of the riboflavin (RF) pathway for TB drug discovery. Authors have used Mtb and Msm and their hypomorphs to test the impact of the RF pathway's gene silencing. The experiments are comprehensive and well-designed, and the results presented are clear. I believe that this study will add knowledge and information on the use of the RF pathway for potential TB drug development.

One of my major comments is that in Fig. 1, the authors have proposed an RF pathway in mycobacteria by inferring from *B. subtilis*. The entire manuscript is based on validating this proposal. Why was this pathway not inferred from mycobacteria? Although, the genetic organizations of these RF biosynthesis genes were confirmed by RT-PCR analysis, I think, the overall aspect of the RF pathway that is stated in Fig.1 has not been confirmed in this study.

Another comment is about content regarding the activation of MAIT cells by riboflavin precursors. Although content related to MAIT cells activation is mentioned in the abstract and indirectly in the result section, no direct experiments have been conducted to show the activation of MAIT cells by metabolites of RF biosynthesis. Authors have only constructed RF pathway hypomorphs and have shown the results of gene silencing. Overemphasis on content MAIT cells is misleading.

In addition, I have a concern regarding the discrepancy between the findings from this study and that by Dey et al (Ref. 54). Dey et al reported that overexpression of ribA2 led to attenuation of Mtb virulence, however, here authors have shown that silencing of ribA2 is bactericidal in both Mtb and Msm. Why is there a difference in the result of this key RF pathway gene from the other study?

Minor comments:

- Fig. 1 should have A and B sub-headings.
- I noticed that there was no effect of ribC silencing in Mtb and Msm (Fig. 2 and Fig. 3), but there was a significant difference in ribC transcripts in Mtb (Fig S5).
- Further, I noticed that the impact of RF pathway gene silencing on the expression of the target/s and other genes is not shown for Mtb.

RESPONSE TO REVIEWERS' COMMENTS: Spectrum03207-23

We thank the reviewers for the constructive comments which we have addressed as outlined below.

REVIEWER 1

1. The authors measure gene expression of RF mutants. However, there are substantial effects on growth of the different mutants - its hard to interpret the qPCRs if the bacteria are in different growth phases. Can the authors perform experiment to measure RF gene expression at different growth phases, maybe this can also explain also the discrepancy of protein vs. mRNA of ribF? The effects of ribA2 on ribG and ribC? ribH on ribF? Also, can they add a non-specific target (as described by Rock et al: <https://www.nature.com/articles/nmicrobiol2016274>) for qPCR, and look at No-ATc and ATc+ to evaluate the general impact of the inserted CRISPR system on gene expression. From fig.S3 and S6 it does seem like the mutants are affected in ability to grow regardless to the presence of ATc.

Response: We have addressed all points, as follows:

- a. We agree that harvesting the bacteria for mRNA vs. protein or metabolite quantification when the cells are at different growth phases could result in some discrepancies. For RNA extraction, the strains were inoculated at an OD of 0.06 and allowed to grow for 24 hours before harvesting the cells whilst for protein or metabolite extraction, which requires a significantly higher biomass, the cells were inoculated at a higher cell density and harvested after 24 hours of ATc treatment. Irrespective of the starting inoculum, we confirmed significant knockdown of target gene expression using *Msm-ribA2* and *ribA2* expression as an exemplar, as shown in the new Fig. S7. It is also important to note that all RNA and protein data that we report included an internal control in that each ATc-treated strain was normalized against itself in the absence of ATc treatment. As a result, we are still able to draw comparisons of the level of transcript amongst strains following gene repression. Since the protein and metabolite extractions were conducted under the same growth conditions, we were able to directly correlate the two.

We further acknowledge that we would not be able to account for differences in mRNA expression vs. protein abundance as these do not always correlate due to post-translational processes that can impact protein modifications and turnover (<https://genomebiology.biomedcentral.com/articles/10.1186/gb-2003-4-9-117> and, [https://www.cell.com/cell/pdf/S0092-8674\(16\)30270-7.pdf](https://www.cell.com/cell/pdf/S0092-8674(16)30270-7.pdf)), factors which are beyond the scope of this study to test. In the case of RibF, we hypothesize that this protein may have high stability once synthesized, resulting in detection of the protein even when the mRNA is depleted. This would also explain why detection of riboflavin in the *ribF* hypomorph is not significantly different when compared with the wild type. We propose that the RibF enzyme is still present and active, hence riboflavin continues to be synthesized even after silencing *ribF* gene expression.

- b. A non-specific targeting sgRNA carrying hypomorph, *Msm_NT* was created and both qPCR and growth kinetics performed. We used the same scrambled sgRNA sequence as described by Bosch *et al.* (PMID: 34297925) to construct this strain. The growth and viability, responsiveness as a function of inoculum size, and expression of RF pathway genes of *Msm_NT* were compared to *Msm_vector*, which only carries the CRISPRi vector backbone. These data are shown as additions to Fig. S3 and Fig. S6 and in the new Fig. S5, respectively. The data shown in Fig. S3 and Fig. S6 confirm that *Msm_NT* is indistinguishable from *Msm_vector* in terms of growth, viability and response to inoculum size. The data in Fig. S5 show significant upregulation of *ribA2* in the presence of ATc in both *Msm_vector* and *Msm_NT*, but no significant impact of ATc on expression of the other pathway genes. Together, these results substantiate the use of *Msm_vector* as an appropriate negative control.

The text has been revised to include the following:

Lines 180-185: “As a further control for potential effects of the CRISPRi vector system on the observed phenotypes, a *Msm* strain carrying a scrambled non-targeting sgRNA (*Msm_NT*) was assessed alongside the vector-only control (*Msm_vector*) and the RF pathway hypomorphs (**Fig. S3**). The lack of impact of ATc on growth of either strain substantiated the use of *Msm_vector* as an appropriate negative control.”

Lines 193-196: “Analysis of expression levels of RF pathway genes in the *Msm_NT* and *Msm_vector* strains revealed an impact of the CRISPRi system on expression of *ribA2*, with both strains showing a ~2-3-fold increase in *ribA2* expression in the presence of ATc (**Fig. S5**). However, expression of the other pathway genes was unaffected.”

Lines 234-237: “This suggested that target gene silencing would still occur under the culture conditions used for proteomic analysis, a conclusion consistent with the ATc-responsiveness of transcript levels of *ribA2* in *Msm_ribA2* when cultures were seeded at varying cell densities (**Fig. S7**).”

- c. The *Msm_vector*, *Msm_ribF* and *Msm_WT* strains had originally been grown in larger culture flasks with greater aeration, which could account for the differences in growth kinetics (as demonstrated by overall higher OD₆₀₀ readings) of these strains compared to the other *Msm* strains. We therefore reassessed the growth and viability of these three strains when cultured under the same conditions as the other strains. Fig. S6 has been revised to include the new data.
2. Can the authors explain better some discrepancies between Fig 2 (growth of the 2-fold dilution colonies) and fig. S3 and S6 (growth curves in broth). Some of these seem to be important. For example, RibG shows partial growth deficiency between 2-fold dilutions. However, RibG in fig.S3 show not growth deficiency over 24h w/wo ATc, while RibG hypomorphs in fig S6 grow more poorly in all dilutions, w/wo ATc. In fig.2, RibH show a greater deficiency in growth, which does reflect in fig.S3 but not in the highest dilution of

fig.S6, where the difference between w/wo ATc seems minor. Maybe perform the growth in broth also with or without RF (as in Fig. 2)?

Response: We have addressed the various points as follows:

- a. We reported that the phenotypes in liquid culture were “broadly similar” to those observed on solid media and made no claim (nor expectation) that the results would be identical. As the reviewer points out, there are some differences in ATc responsiveness of growth in liquid vs. solid media, e.g. for *Msm_ribG*. (We have made analogous observations for other CRISPRi hypomorphs in other studies in our lab.) The two growth phenotyping assays are not directly comparable: the liquid culture assay is run over 24 h, sampling every 3 h, whereas the spotting assay scores growth after 2-3 days (as an endpoint assay). Moreover, in the liquid assay *Msm_ribG* did show evidence of retarded growth and impaired survival at the 6, 9, 21 and 24 h time points although the effect was subtle.
 - b. The highest dilution culture of *Msm_ribH* assessed in Fig. S6 (red, starting OD = 0.004) shows significant growth impairment at the 15, 18 and 21-h time points. The SD for the 24-h time point is high, which suggests significant growth at that time point. Based on the unexpectedly rapid rise in growth of one of the biological replicates between the 21- and 24-h time points, we speculate that this strain had undergone spontaneous loss of ATc responsiveness via pseudoreversion, which would account for this observation.
 - c. In Fig. S8, we showed that growth *Msm_ribH* hypomorph in broth culture was only partially rescuable by riboflavin, requiring >1.3 mM riboflavin for partial rescue. Therefore, the riboflavin rescue phenotype in liquid culture parallels that observed on solid media.
3. Can the authors address the differences in growth of hypomorph RibH in Fig. 2 and deletion mutant Δ RibH in Fig. 6, especially in RF supplemented plates. Why does complete deletion of this gene, which is considered essential, allow growth in RF? Is it possible that the gRNA targets both *ribH1* and *ribH2* (*msm6598*)? Can qPCR be done to validate this?

Response: We propose that the inability of RF to rescue the depletion of RibH in this hypomorph may be due to an off-target effect. In other words, in addition to silencing *ribH*, the sgRNA may be binding to another essential gene, affecting an unrelated pathway and thereby preventing complete rescue with RF supplementation. We have included new qPCR data in Fig. S11 which show that whereas *ribH* (“*ribH1*”) expression is knocked down in the *Msm_ribH* hypomorph, the transcription of *MSMEG_6598* (“*ribH2*”) is unaffected by silencing of *ribH* (“*ribH1*”) in this strain.

*New text added to the manuscript, lines 295-297: “Furthermore, we verified that in *Msm_ribH*, the sgRNA knocked down the expression of *ribH* while expression of the second lumazine synthase-encoding gene, *MSMEG_6598*, was unaffected (Fig. S11).”*

4. There seems to be some interesting differences between mRNA, protein and metabolite levels (RF). Can the authors better explain how these fit with the RF synthesis pathway. For example, KD of RibF showed reduced mRNA (fig. 3) but not protein (fig. 4). However, if ribF is indeed KD it would be expected to result in increase of RF (fig. 5), as there should be no enzyme to convert RF.

Response. This comment has been partly addressed above under point 1 above. Protein abundance and mRNA level trends show concordance for most of the genes targeted, i.e., in cases where the transcription is reduced as shown by qPCR, proteomics data also shows a reduction in protein levels in spite of the difference in conditions used to generate the data. The exception to this is RibF where we propose that despite significant reduction in the level of transcript, residual RibF enzyme is present, allowing ongoing synthesis and detection of riboflavin.

5. Can the authors explain in more detail how were the guides for each gene chosen?

Response. A comprehensive list of computationally designed sgRNAs (with different potencies) targeting all genes in Msm and Mtb described by Bosch *et al.* (PMID: 34297925) were selected to target genes of interest in the pathway explored in this study. Of these, we selected several sgRNAs to target each gene and experimentally tested their potency using an ATc gradient in a resazurin-based assay (data not shown). From these data, we selected the sgRNAs demonstrating the most potent growth inhibition to take forward for further characterization.

New text added to the manuscript, lines 163-167: “For each gene, 5-15 sgRNAs with varying predicted efficiencies of knockdown were used (Fig. S2). All resulting hypomorphs were screened for ATc responsiveness over a range of ATc concentrations (0.1 - 200 ng/ml; data not shown). For each gene, the most ATc-responsive hypomorph (or the one with the lowest PAM score, in the case of *ribC*) was selected for further characterization (Fig. S2).”

6. For Mtb - in Fig. S5 - will it be possible to look at how specific is the targeting on other genes in the pathways? For example, it will be interesting to see if *ribA2* has also other off-targets like in *msm*?

Response: This analysis was performed, and the data included in a new figure (Fig. 4).

New text added to the manuscript, lines 214-222: “The impact of knockdown of individual RF pathway genes on expression of pathway genes was then assessed in Mtb (Fig. 4). Consistent with the operonic arrangement, transcriptional silencing of *ribA2* led to knockdown of *ribA2* as well as *ribH*, but unlike Msm, *ribA2* silencing in Mtb did not affect expression of other genes in the pathway. Silencing of *ribA2*, *ribH* or *ribC* was specific, showing highly significant knockdown of the cognate gene. Although ATc-dependent knockdown of *ribG* or *ribF* did not reach statistical significance in their corresponding hypomorphs under the conditions tested, the survival of Mtb_*ribG* and Mtb_*ribF* was nonetheless impaired under the conditions used to assess growth and survival (Fig. S4). Silencing of *ribF* also led to a significant upregulation of both *ribG* and *ribH*.”

7. In Fig 1, add location of RibA1 and RibH2 (MRMEG_5698) in msm, as they are referred to in the papers and fig. 6. Also, a step is missing to describe RibG bi-functionality - ARPP conversion to ArPP, which is generated to ArP by phosphatase.

Response. As suggested, MSMEG_6598 has been added to the RF pathway figure (Fig. 1). Since the monofunctional RibA1 cannot compensate for the loss of RibA2 we mentioned this gene instead in the revised figure legend. The additional step mediated by RibG has also been added to the pathway figure.

REVIEWER 2

1. One of my major comments is that in Fig. 1, the authors have proposed an RF pathway in mycobacteria by inferring from *B. subtilis*. The entire manuscript is based on validating this proposal. Why was this pathway not inferred from mycobacteria? Although, the genetic organizations of these RF biosynthesis genes were confirmed by RT-PCR analysis, I think, the overall aspect of the RF pathway that is stated in Fig.1 has not been confirmed in this study.

Response. The riboflavin biosynthesis pathway has been best characterized in *Bacillus* sp., used in industry as a model organism to overproduce riboflavin (see link.springer.com/article/10.1007/BF02931951; link.springer.com/article/10.1007/BF02931951; link.springer.com/article/10.1007/s004380050393). We used this model pathway as a reference since it depicts the complete sequence of biosynthetic events needed to produce riboflavin. We did not use the *Bacillus* sp. genes from the RF pathway to bioinformatically identify mycobacterial homologs; with the exception of the phosphatase, the homologs of the other genes were readily identified from their functional annotation in the Mycobrowser database (<https://mycobrowser.epfl.ch/>). We recognize that we did not communicate this clearly in the manuscript and have therefore removed the statement that the pathway was inferred from *Bacillus* sp.

2. Another comment is about content regarding the activation of MAIT cells by riboflavin precursors. Although content related to MAIT cells activation is mentioned in the abstract and indirectly in the result section, no direct experiments have been conducted to show the activation of MAIT cells by metabolites of RF biosynthesis. Authors have only constructed RF pathway hypomorphs and have shown the results of gene silencing. Overemphasis on content MAIT cells is misleading.

Response: This point is well taken. The RF pathway hypomorphs were generated in this study as a tool for *future* studies to investigate the link between RF pathway metabolites and MAIT cell activity. We have clarified this point by removing reference to MAIT cell recognition in the Introduction and stressing in the Abstract and Introduction that the hypomorphs represent a valuable resource for *future* studies.

3. In addition, I have a concern regarding the discrepancy between the findings from this study and that by Dey et al (Ref. 54). Dey et al reported that overexpression of ribA2 led to attenuation of Mtb virulence, however, here authors have shown that silencing of ribA2 is bactericidal in both Mtb and Msm. Why is there a difference in the result of this key RF pathway gene from the other study?

Response. In this study we showed that the depletion of RibA2 in Mtb was bactericidal *in vitro*. Dey *et al.* (ref. 55) demonstrated that overexpression of *ribA2* did not affect the growth of Mtb *in vitro* but attenuated Mtb virulence which they attributed to an increased production of MAIT cell activating ligands which allowed the infection to be effectively contained. While our manuscript was under review, other authors showed that transposon mutants of Mtb in *ribA2* and *ribG* show significant defects in mice. This is consistent with our *in vitro* results and confirms that RF biosynthesis is essential for Mtb growth *in vivo*. We have cited this new paper as a new reference no. 54.

New text added to the manuscript, lines 360-361: “This conclusion is consistent with the profound fitness defects of transposon mutants of Mtb in *ribA2* and *ribG* reported recently in mice (54).”

4. Fig. 1 should have A and B sub-headings.

Response. Done

5. I noticed that there was no effect of ribC silencing in Mtb and Msm (Fig. 2 and Fig. 3), but there was a significant difference in ribC transcripts in Mtb (Fig S5).

Response: The apparent disconnect between *ribC* silencing, RibC protein depletion and growth phenotype in Msm was noted, as described in lines 238-241: “The lack of an effect of *ribC* silencing on Msm growth (Fig. S36) under conditions resulting in significant depletion of RibC protein (Fig. 5A) is consistent with the relative invulnerability of RibC in this organism (Table S1).” We speculate that the reason underlying the disconnect between *ribC* transcript knockdown and growth/survival impairment in Mtb is analogous to that proposed for Msm but are unable to confirm this in the absence of a targeted proteomic analysis of the level of RibC protein in Mtb, which we respectfully submit falls beyond the scope of the current manuscript.

6. Further, I noticed that the impact of RF pathway gene silencing on the expression of the target/s and other genes is not shown for Mtb.

Response. This has been done and the results reported in a new figure (Fig. 4), as described in the response to point 6 raised by Reviewer 1.

Re: Spectrum03207-23R1 (Modulation of riboflavin biosynthesis and utilization in mycobacteria)

Dear Dr. Valerie Mizrahi:

Thank you for making critical revisions suggested for your manuscript.

Your manuscript has been accepted, and I am forwarding it to the ASM production staff for publication. Your paper will first be checked to make sure all elements meet the technical requirements. ASM staff will contact you if anything needs to be revised before copyediting and production can begin. Otherwise, you will be notified when your proofs are ready to be viewed.

Sincerely,
Gyanu Lamichhane
Editor
Microbiology Spectrum